# Comparison of the Hemocompatibility of an Axial and a Centrifugal Left Ventricular Assist Device in an In Vitro Test Circuit

**DOI:** 10.3390/jcm11123431

**Published:** 2022-06-15

**Authors:** Patrick Borchers, Patrick Winnersbach, Sandra Kraemer, Christian Beckers, Eva Miriam Buhl, Steffen Leonhardt, Rolf Rossaint, Marian Walter, Thomas Breuer, Christian Bleilevens

**Affiliations:** 1Medical Information Technology, RWTH Aachen University, Pauwelsstraße 20, 52074 Aachen, Germany; borchers@hia.rwth-aachen.de (P.B.); leonhardt@hia.rwth-aachen.de (S.L.); walter@hia.rwth-aachen.de (M.W.); 2Department of Anesthesiology, University Hospital RWTH Aachen, Pauwelsstraße 30, 52074 Aachen, Germany; pwinnersbach@ukaachen.de (P.W.); rrossaint@ukaachen.de (R.R.); 3Department of Intensive Care and Intermediate Care, University Hospital RWTH Aachen, Pauwelsstraße 30, 52074 Aachen, Germany; skraemer@ukaachen.de (S.K.); cbeckers@ukaachen.de (C.B.); 4Institute of Pathology, Electron Microscopy Facility, RWTH Aachen University, Pauwelsstraße 30, 52074 Aachen, Germany; ebuhl@ukaachen.de

**Keywords:** left ventricular assist device, in vitro test circuit, Sputnik, HeartMate 3, hemocompatibility, hemolysis, coagulation, platelet activation, extracellular vesicles

## Abstract

Background: Hemocompatibility of left ventricular assist devices is essential for preventing adverse events. In this study, we compared the hemocompatibility of an axial-flow (Sputnik) to a centrifugal-flow (HeartMate 3) pump. Methods: Both pumps were integrated into identical in vitro test circuits, each filled with 75 mL heparinized human blood of the same donor. During each experiment (n = 7), the pumps were operated with equal flow for six hours. Blood sampling and analysis were performed on a regular schedule. The analytes were indicators of hemolysis, coagulation activation, platelet count and activation, as well as extracellular vesicles. Results: Sputnik induced higher hemolysis compared to the HeartMate 3 after 360 min. Furthermore, platelet activation was higher for Sputnik after 120 min onward. In the HeartMate 3 circuit, the platelet count was reduced within the first hour. Furthermore, Sputnik triggered a more pronounced increase in extracellular vesicles, a potential trigger for adverse events in left ventricular assist device application. Activation of coagulation showed a time-dependent increase, with no differences between both groups. Conclusions: This experimental study confirms the hypothesis that axial-flow pumps may induce stronger hemolysis compared to centrifugal pumps, coming along with larger amounts of circulating extracellular vesicles and a stronger PLT activation.

## 1. Introduction

Due to the worldwide shortage of donor organs, including donor hearts, left ventricular assist devices (LVADs) are increasingly used for the treatment of end-stage heart failure patients [1]. However, LVADs can induce blood trauma due to mechanical forces, heat or due to blood contact with the foreign surface material of the pump [2]. Adequate hemocompatibility of LVADs is a fundamental necessity to prevent adverse events, such as anemia, thromboembolic events, gastrointestinal bleeding or stroke [3]. To evaluate the hemocompatibility of LVADs, various parameters need to be investigated.

Hemolysis (the lysis of red blood cells (RBCs)) is one of the most crucial parameters for assessing the hemocompatibility of blood pumps [4]. LVAD-induced hemolysis is strongly affected by the pump-induced shear stress and the residence time of blood cells in high shear stress regions [5]. The extent of shear stress caused by LVADs is dependent on its design type [6]. High shear stress regions are more prominent in continuous-flow rotary LVADs compared to pulsatile hydraulic LVADs, even though their overall number of adverse events is lower [7,8,9]. Hemolysis causes the intracellular hemoglobin of RBCs to be released into the blood plasma. Therefore, the amount of plasma-free hemoglobin is a classical marker to assess hemolysis [10].

As another important parameter to elucidate hemocompatibility, the activation of coagulation and platelets (PLTs), as well as general blood cell counts, are used [6]. Like the white blood cell (WBC) count, the amount of activated PLTs serves as an indicator of increased inflammation and coagulation [11]. In particular, PLT activation is known to be induced by high shear stress, indicated by the expression of distinct markers on the PLT surface, inducing pro inflammatory and procoagulatory cascades [12]. Additionally, PLT count decrease is a clinically well-known reason for accompanying bleeding complications [13,14]; thus, PLTs play a central and crucial role in the determination of hemocompatibility. Activation of coagulation is an indicator of clot formation and can be determined, among others, via thrombin/antithrombin complexes [15].

As a novel biomarker of interest to evaluate the hemocompatibility of LVADs, extracellular vesicles (EVs) are focused on in recent publications [16,17]. EVs are small lipid structures (<1 µm) released by all types of blood cells. Generally, EVs contain cargo, such as proteins [18] and miRNAs [19], which can mediate physiological changes in their target cells—beneficial or detrimental. LVADs are known to increase circulating EVs in the blood [16,17,20,21]. Whether EVs released due to LVADs could be biomarkers of adverse events needs to be investigated in more detail and, therefore, is included in this study.

The Sputnik pump is an axial-flow LVAD and has been implanted around 50 times by the beginning of 2020 [10]. Romanova et al. experimentally compared the pump-induced hemolysis levels of the Sputnik LVAD and its new generation under development [22]. However, in terms of good scientific practice, the Sputnik LVAD should also be compared to a state-of-the-art LVAD, which is intended for a similar application and flow rate [2]. The axial-flow LVAD HeartMate 2 would be an obvious choice for comparison, as it has already been implanted more than 27,000 times [23]. However, the HeartMate 3 centrifugal pump, as the successor of the HeartMate 2, can be regarded as the novel state-of-the-art LVAD according to the manufacturer [24]. This is justified by lower event-free survival after 2 years of the HeartMate 3 compared to the HeartMate 2 [25]. Furthermore, the HeartMate 3 received approval for the CE mark in 2015 [26] and has been evaluated in long-term studies in 2200 patients [27,28]. Therefore, the HeartMate 3 LVAD was chosen as the comparative device.

Especially in the Asia-Pacific region, there is a high interest in lower-cost alternatives for established LVADs to offer LVAD therapy to a larger patient population [29]. One potential alternative device is the Sputnik LVAD, which is in clinical use in Russia, but data on clinical performance or hemocompatibility properties are lacking.

The aim of this study was to compare the hemocompatibility of the Sputnik LVAD with a gold standard LVAD, the HeartMate 3 in this case, in an in vitro and short-term experimental study for the first time. Therefore, human whole blood was circulated in two identical test circuits, and several parameters were determined to assess the hemocompatibility of both LVADs. However, this study solely represents the first insight on an experimental level.

Furthermore, axial-flow LVADs are typically associated with higher pump-induced hemolysis compared to centrifugal LVADs in in vitro experiments [3,5,7]. This hypothesis was further investigated in this study as well.

## 2. Materials and Methods

### 2.1. Technical Features of Pumps under Investigation

The Sputnik LVAD [30,31] is an axial-flow LVAD with blood flowing in parallel to the impeller axis (Figure 1A). It was developed by the National Research University of Electronic Technology (MIET) and others in Russia.

The Sputnik pump contains an impeller with four blades, which is rotated via coils inside the stator. The impeller has a diameter of 15.6 mm, and the flow channel has a diameter of 16 mm, which results in a flow gap of 0.2 mm. The impeller is suspended on two mechanical needle bearings on the opposite sides of the pump. The bearing material consists of a cobalt–chromium–molybdenum alloy. A flow straightener with four blades at the pump input minimizes the eddy flow before entering the impeller. Furthermore, the pump output is equipped with a diffusor consisting of three blades.

The Sputnik pump is connected to the left ventricle via an inflow cannula and to the aortic arc via an outflow cannula, felt ferrule and vascular prosthesis (sewed to the ascending aorta).

The HeartMate 3 LVAD [9,26], on the other hand, is a centrifugal pump, which radially ejects blood entering axially to the impeller axis. It was originally designed by Thoratec Corp., which was acquired by Abbott (North Chicago, IL, USA) in 2017 [33].

As shown in Figure 1B, it consists of a fully magnetically levitated rotor with wide blood flow gaps for increased durability and minimized shear stress. This is achieved by passive rotor magnets and electromagnetic stator coils for drive and levitation. Blood flow gaps are 0.5 mm on the side of the rotor and 1 mm on the top and bottom of the pump, which is much larger compared to the Sputnik LVAD. Therefore, the stasis and damage of blood components can be assumed to be lower. The levitation is controlled via distance sensors, and the velocity is controlled via hall sensors. To further improve hemocompatibility, the pump is textured by a sintered titanium microsphere surface. This induces the adhesion of circulating cells and the development of a stable biological lining to reduce anticoagulation requirements and thromboembolism [34].

A special feature of the HeartMate 3 is the artificial pulse it produces 30 times per minute asynchronously from heart activity. During each artificial pulse, the pump speed is decreased by 2000 rpm for 0.15 s and subsequently increased by 4000 rpm for 0.2 s. However, the artificial pulse is only generated for speed levels greater than 4000 rpm.

For HeartMate 3 implantation, the 20 mm inflow cannula is positioned in the apex of the left ventricle. It is secured in place with an apical cuff and a titanium-locking ring. A 14 mm sealed outflow graft with bend relief is anastomosed to the ascending aorta.

Table 1 summarizes the key features of the Sputnik and the HeartMate 3 pumps, respectively. Both pumps come with portable control units, including battery supply, which are connected to the LVAD via a percutaneous cable passing through the patient’s skin. The corresponding control units for Sputnik and HeartMate 3 were utilized in this study.

### 2.2. Test Circuit Design

For each experiment, two similar test circuits, consisting of an LVAD (Sputnik/HeartMate 3), a reservoir (Small pouch, part of CompoSelect^®^ RCC, Fresenius Kabi AG, Bad Homburg, Germany) and connecting PVC tubes (1/200 and 1/400 RAUMEDIC AG, Münchberg, Germany) were assembled to form a closed loop of 75 mL total volume (Figure 2). The test circuits were constructed in the same way, except for the installed LVAD. Except for the miniaturization, the test circuits were designed according to the “Standard Practice for Assessment of Hemolysis in Continuous Flow Blood Pumps” (ASTM F1841). The reservoirs are mandatory to achieve enough blood volume within the circuits for multiple blood withdrawals during the experiments. For better handling, the whole test circuits were fixed in an upright position to a vertical panel.

Blood flow was measured via flow sensors (BioProTTTM Clamp-OnTM Transducer, em-tec GmbH, Finning, Germany) connected to the PVC tubing. Pressure sensors (Pressure Probe Xtrans, CODAN pvb Critical Care Inc., Forstinning, Germany) were connected before and behind the LVADs via Luer connectors for continuous pressure monitoring within the systems.

A steady temperature of 37 °C within the circuits was assured by using a heating hood (Certomat HK, Labexcahnge.com, Burladingen, Germany). The specific temperature of each test circuit was measured at the reservoir via a standard temperature probe (DATEX AS/3, GE Healthcare, Solingen, Germany).

### 2.3. Circulation and Hemodynamics

In this miniaturized experimental setting, it is not possible to run both devices at their optimal speed, as the resulting flow patterns would differ between the devices. Therefore, an experimental setting was chosen, which enables optimal comparison between the devices, regardless of the optimal conditions for the devices. Hence, the automatically generated pulsatile waveform, which is the optimal working mode of the HeartMate 3, was avoided to enable a fair comparison with the non-pulsatile flow of the Sputnik. This was achieved by limiting the speed of the HeartMate 3 to 4000 rpm, which results in a constant pump flow of about 250 mL/min. This is equal to a circulation of the whole blood in the circuit of 3.3 turnovers per minute, and the speed of the Sputnik pump was adjusted to generate 250 mL/min likewise. In relation to a clinical scenario, with 5 L/min, which corresponds to one turnover of the whole blood amount of an adult patient, our setting was 3–4 times higher. In relation to the proposed worst-case clinical operation conditions of 5 L/min proposed by the ASTM F1841 for a 450 mL test circuit, which equals 11.1 turnovers per minute of the whole blood volume, we were exactly in between these two settings. Repeated blood sampling during the experiments further decreased the blood volume inside the miniaturized test loops to 60 mL after 240 min. Therefore, blood circulation rates within the loops were 3.33 turnovers per minute at the beginning and 4.17 turnovers per minute after 240 min.

The offset pressure was regulated to ≥75 mmHg via a custom-made clamp module to adjust the pressure inside the reservoir bag. The clamp modules were readjusted after every blood sampling due to volume and subsequent pressure loss. The overall resistance of the loop resulted in head pressures of 40–50 mmHg.

### 2.4. Blood Donation and Experimental Groups

For each experiment, 150 mL human blood from a single donor was withdrawn into three 50 mL syringes, primed with 3.75 IU/mL of heparin (LEO Pharma A/S, Ballerup, Denmark). The blood was withdrawn from the median cubital vein of volunteers after informed consent and approval of the ethical committee of the University Hospital of the RWTH Aachen (file no EK134/20). In that way, blood donations from 7 healthy volunteers (4 male, 3 female) were used for seven experiments. Each donation was split into two equal portions (75 mL) for simultaneous operation of the test circuits and direct comparison of two experimental groups (Sputnik/HeartMate 3). Shortly after blood withdrawal, both test circuits were filled carefully, avoiding entrapped air, and started simultaneously.

### 2.5. Blood Sampling and Analysis

Directly after withdrawal, blood was initially analyzed to determine the patient’s baseline levels. Subsequently, blood was sampled (3 mL per sample) from the loop after 5 min and after 1, 2, 3, 4 and 6 h, respectively. Due to the limited sample volume, no sample was collected after 5 h. Immediately after each sampling, a hemogram was measured on an automated cell counter (MEK-6550K, Nihon Khoden Inc., Rosbach, Germany), and blood gas analysis was performed on an ABL800 analyzer (Radiometer Inc., Fichtenhain, Germany). Blood for subsequent measurements was collected into citrate collection tubes (S-Monovette, Sarstedt Inc., Nümbrecht, Germany). To obtain plasma, the blood was centrifuged at 2000× *g* for 10 min at room temperature, then aliquoted and stored at −80 °C. The levels of thrombin/antithrombin complex (TAT) in the plasma were determined from these samples in accordance with the manufacturer’s instructions by an enzyme immunoassay using a TAT ELISA Kit (Thrombin/Antithrombin Complex (TAT) ELISA Kit (Human), SEA831Hu, Cloud-Clone Corp, Katy, TX, USA).

### 2.6. Fluorescence-Activated Cell Sorting

The ratio of activated platelets was determined by detecting the platelets’ cell surface expression of P-selectin (CD62P+). An amount of 20 µL citrated whole blood from the test circuit was diluted 1:50 and incubated with fluorescein isothiocyanate-conjugated and phycoerythrin-conjugated CD61 and CD62P (MCA2263F, MCA2418PE, BioRad Inc., Feldkirchen, Germany) or their isotype controls (MCA928F, MCA 928PE, BioRad Inc., Feldkirchen, Germany) at room temperature for 15 min. The platelets were fixed (Cell Fix; BD Biosciences, Heidelberg, Germany) after staining and subsequently analyzed using a flow cytometer (BD FACSCanto, BD Biosciences, San Jose, CA, USA).

### 2.7. Isolation of Extracellular Vesicles

Extracellular vesicles were isolated by precipitation, using the commercial ExoQuick Ultra Kit (System Biosciences, Palo Alto, CA, USA). Plasma samples were centrifuged at 4 °C for 10 min at 12,000× *g*, and 200 µL supernatant was transferred into a new reaction tube. The isolation of EVs was performed according to the manufacturer’s protocol. Briefly, the appropriate volume of ExoQuick was added to the plasma samples, mixed by inverting, and incubated for 30 min at 4 °C. The mixture was centrifuged at 4 °C for 10 min at 3000× *g*. Supernatants were aspirated and discarded without disturbing the precipitated EVs. The pellets were resuspended in Buffer B and added to the resin of prepared columns and mixed at RT for 5 min on a rotation shaker. Columns were centrifuged at 4 °C for 30 s at 1000× *g* to obtain the purified EVs. EVs were aliquoted and stored at −80 °C until further analysis.

### 2.8. Western Blot Analysis

Antibodies for the detection of specific proteins, which are associated with EVs and specific for platelets or red blood cells, were purchased from BD Biosciences (Flotillin-1: BD610821, BD Biosciences inc., Heidelberg, Germany), Cell Signaling Technologies (CD9: #13403, Integrin a2b: #13807, Cell Signaling Technologies, Danvers, MA, USA), Abcam (Band3/AE1: ab108414, Abcam, Cambridge, UK) and Bio-Techne (ApoB100: AF3260, Bio-Techne, Wiesbaden, Germany). As secondary antibodies, anti-mouse- (#Na931V, GE Healthcare, Munich, Germany), anti-rabbit- (#7074, Cell Signaling Technologies, Danvers, MA, USA) and anti-goat antibodies (sc-2354, Santa Cruz Biotechnology, Heidelberg, Germany) were used.

An amount of 40 µL of each sample, respectively, 20 µL for ApoB100 detection were separated by gel electrophoresis using 12% (4–20% for ApoB100) Acrylamide Stainfree-gels (# 1610175/#4561095, BioRad, Feldkirchen, Germany), as described previously [37]. The samples for the detection of Band3, Integrin a2b, ApoB100, Flotellin-1, CD63 and CD9 were prepared as described previously [37]. The normalization of the single EV markers was performed using the total protein amount of the corresponding sample, visualized within the Acrylamide Gel after electrophoresis and prior to the blotting procedure by the Stainfree technology, as described previously [3].

Subsequently to gel electrophoresis, the separated proteins were transferred to a PVDF membrane (#1704150, BioRad, Feldkirchen, Germany), blocked, washed and incubated [35]. Chemiluminescent signals were detected after incubation of the membrane with SuperSignal West Femto Substrate (34094, Thermo Fisher Scientific, Langerwehe, Germany), using the ChemiDoc Imaging System (BioRad, Karlsruhe, Germany). Densitometrical analysis of the protein bands’ intensity was performed using the image lab Software (BioRad) and displayed as relative units normalized to the stain-free images and referenced to baseline values.

### 2.9. Nanoparticle Tracking Analysis (NTA)

The size and concentration of isolated EVs were quantified by NTA using the NanoSight NS300 (Malvern Panalytical Ltd., Malvern, UK) equipped with a sCMOS camera and a Blue488 laser module. All samples were diluted 1:50–1:100 in DPBS (D8662, Sigma Aldrich, Taufkirchen, Germany) to a final volume of 1 mL. Capture settings were used according to the manufacturer’s manual; camera focus was adjusted by autofocus; and the camera level was set to 14. For each measurement, three 60 s videos were captured. The videos were analyzed by the NanoSight Software NTA 3.2 Dev Build 3.2.16 with a detection threshold of 7.

### 2.10. MiRNA Isolation and Quantification

MiRNA was isolated by spin column purification, using the commercial miRNeasy Advanced Kit (217204, Qiagen, Hilden, Germany). The isolation was performed according to the manufacturer’s protocol. Briefly, 200 µL of purified EVs were lysed with 60 µL Buffer RPL, vortexed for 5 s and incubated for 3 min at RT. After lysis, 60 µL Buffer RPP was added, vortexed for 20 s, incubated for 3 min and centrifuged for 3 min at 12,000× *g* to precipitate proteins. Afterward, the supernatant was transferred to a new reaction tube and mixed with isopropanol. The mixture was transferred to a column and centrifuged for 15 s at 8000× *g*, followed by subsequent washing steps. RNA was eluted with RNase-free water. MiRNA was quantified using a eukaryotic total RNA Pico chipset (#5067-1513 Agilent technologies, Santa Clara, CA, USA).

### 2.11. Transmission Electron Microscopy (TEM)

Isolated EVs were incubated for 5 min on glow discharged formvar carbon-coated grids (Nickel Grid 200 mesh, Electron Microscopy Sciences, Hatfield, PA, USA). The grids were washed three times with H_2_O. Negative staining was performed with 0.5% uranyl acetate (Electron Microscopy Sciences, Hatfield, PA, USA). Excess liquid was removed using filter paper. The grids were air dried for 10 min. Samples were imaged using an LEO 906 EM transmission electron microscope (Zeiss, Oberkochen, Germany), operated at an acceleration voltage of 60 kV.

### 2.12. Statistical Analysis

All data, except for EVs, are presented as mean ± SEM. The normality of the data was checked by a Shapiro–Wilk normality test. A two-way ANOVA with Sidak correction for multiple comparisons and a confidence interval of 95% was performed to verify differences at single time points between the two paired groups (Sputnik/HeartMate 3), using the GraphPad Prism software (GraphPad Prism version 9.2.0 for MacOS, GraphPad Software, San Diego, CA, USA). If the calculated *p*-value was <0.05, the results were regarded as significantly different. GraphPad Prism was also used to design the graphs.

## 3. Results

### 3.1. Operating Parameters

Figure 3 compares the operating conditions of the Sputnik and HeartMate 3 test circuits shortly before each sample collection. The flow (Figure 3B) follows the preset of constant 250 mL/min throughout the experiments, with minimal variations and no differences between the groups. However, the pump speed (Figure 3A) needed to generate equal flow in both test circuits was significantly higher at every time instant in the Sputnik group (5000 rpm) in comparison to the HeartMate 3 (3750 rpm) group. Furthermore, the head pressure (Figure 3C) reveals no significant differences between the groups.

The temperature (Figure 3D) strongly increased over the first hour in both groups. This occurred due to the initial heating of the blood inside the heating hood. Comparing both groups, the temperature was significantly higher in the Sputnik group after 60 min. This significant difference could be explained by the stronger self-heating of the Sputnik pump due to higher pump speed and heat generation in its mechanical bearings [3].

### 3.2. Indicators of Hemolysis

The concentration of fHGB (Figure 4A) and potassium (Figure 4B) increased gradually during circulation in the Sputnik group. Potassium was statistically significantly higher after 180 min, whereas fHGB was statistically significantly higher after 360 min in the Sputnik group compared to the HeartMate 3 group. As fHGB and Potassium are indicators of hemolysis, these results seem to indicate increased hemolysis in the Sputnik group toward the end of the experiments in comparison to the HeartMate 3 group.

### 3.3. Platelet Count and Activation

PLT activation (Figure 5B) in the HeartMate 3 group showed an increase after 5 min and afterward declined to baseline levels. The PLT activation in the Sputnik group showed a statistically significant increase starting at 120 min in comparison to the HeartMate 3 group. Despite this PLT activation, the PLT count (Figure 5A) in the Sputnik group persisted at baseline levels. In sum, we saw a higher activation of PLTs by the impeller of the Sputnik pump, without relevant clot formation or PLT deposition.

Platelet count decreased in the HeartMate 3 test circuit from 244 ± 49 × 10^3^/μL to 177 ± 67 × 10^3^/μL after 60 min of circulation. In the further course, the PLT count in the HeartMate 3 group remained lower in comparison to the Sputnik group. For the Sputnik group, PLT remained almost constant all the time.

### 3.4. Coagulation Activation

Coagulation activation was closely evaluated via measurement of the thrombin/antithrombin complex (TAT) concentration. TAT concentration is depicted in Figure 5C, and it increased statistically significantly over the time of the experiment in both groups. However, no statistically significant differences in TAT concentrations were observed between the groups.

### 3.5. Cell Counter

Baseline white blood count WBC was 5.4 ± 2.4 × 10^3^/μL. WBC showed a slight decrease until the end of the experiments without differences between the groups (Figure 6C).

The mean total hemoglobin (HGB) concentration (Figure 6A) of the blood donation was 14.2 ± 1.5 g/dL. HGB in the HeartMate 3 group increased slightly after 60 min, whereas HGB remained at baseline levels in the Sputnik group. The increase in the HeartMate 3 group was statistically significant in comparison to the Sputnik group at 180 and 240 min.

### 3.6. Blood Gas Analysis

Glucose (Figure 6F) and pH (Figure 6C) declined gradually in both groups during the experiments. Mean lactate (Figure 6D) increased gradually from 1.3 ± 0.0 (baseline) to 10.8 ± 0.4 at the end of the experiments. There were no differences between the groups. Mean calcium levels showed an initial drop, which was stronger in the Sputnik group, and an increase until the end of experiments in both groups.

### 3.7. Release of Extracellular Vesicles

As another potential damage marker, we isolated EVs from plasma samples at baseline and after 6 h of circulation by precipitation. The quality was assessed by TEM (Figure 7A) and revealed small vesicular structures in the baseline sample with a size <100 nm. However, in the Sputnik and HeartMate 3 samples, visualization of EVs was difficult due to the excessive amount of cell debris, most likely derived from hemolysis. (Figure 7B). Since EVs are known to carry miRNAs, total miRNA content was quantified.

Using a eukaryotic total RNA Pico chip, we could detect a statistically significant increase in miRNA content in the Sputnik EVs (483.30 ± 113.60 pg/µL) compared to baseline (116.20 ± 27.31 pg/µL) and HeartMate 3 EVs (283.8 ± 114.30 pg/µL) (Figure 7B).

EVs were further analyzed by SDS-PAGE and Western blot using antibodies for typical EV markers (Flotilin-1, CD9) (Figure 7C–E), as well as a non-vesicular marker (Apolipoprotein B) (Appendix A). A drastic increase in typical EV proteins was detected in Sputnik and HeartMate 3 EVs (Figure 7C–E). Flotilin-1 was statistically significantly increased in both circulation settings (Figure 7D) compared to baseline EVs, whereas CD9 was only increased in Sputnik EVs (Figure 7E).

Since all blood cells can secrete EVs, we wanted to identify the origin of our isolated EVs. Therefore, we probed the membrane for band 3 anion transport protein, also known as anion exchanger 1 (AE1), which is found in the erythrocyte membrane. As a marker of platelets, we used CD41, which plays a crucial role in coagulation (Figure 8). For Band 3/AE1, no significant differences were detected between baseline and Sputnik or HeartMate 3 EVs, which indicates that erythrocytes are not the parental cells. However, we could detect a statistically significant increase in CD41 in the Sputnik EVs and, to a lesser (not statistically significant) extent, in the HeartMate 3 EVs.

## 4. Discussion

In this study, in vitro test circuits were used to compare two different LVADs (Sputnik/HeartMate 3) in an equal flow scenario, regarding their hemocompatibility and operational parameters, with special attention to their impact on PLTs, coagulation and hemolysis, inspired by Zayat et al. [38]. Therefore, heparinized blood from healthy human volunteers was circulated in two similar test circuits for six hours.

The main significant findings are stronger hemolysis, greater PLT activation and pronounced EV release in the Sputnik group in comparison to the HeartMate 3 group. The HeartMate 3 LVAD induced a significant PLT count decrease in comparison to the Sputnik LVAD. Both LVADs induced significant, time-dependent activation of coagulation. Further assessed parameters do not indicate differences regarding hemocompatibility between the pumps.

### 4.1. Hemolysis

This study reveals that the Sputnik axial-flow LVAD causes more hemolysis, indicated by a time-dependent increase in fHGB and potassium, compared to the HeartMate 3 centrifugal-flow LVAD. The tendency of axial-flow LVADs to cause more hemolysis than centrifugal LVADs is in accordance with previous publications [3,5,7]. For example, Bourque et al. [39] compared the hemolysis levels of the HeartMate 3 to the HeartMate 2 pump. The corresponding experiments according to the ASTM F1841 demonstrate lower hemolysis for the HeartMate 3 compared to the HeartMate 2 for pump flow levels of 2 L/min, 5 L/min and 10 L/min, respectively. A pump flow of 2 L/min in an ASTM loop with 450 mL filling volume corresponds to a blood circulation rate of 4.44 turnovers per minute, which is only slightly higher compared to the circulation rate of 3.33–4.17 turnovers per minute in our study. In the study of Bourque et al. [39], the fHBG of the HeartMate 2 loop increases by 92 mg/dL over 6 h, whereas the fHBG of the HeartMate 3 loop only increases by 38 mg/dL. In our investigation, the HeartMate 3 pump induced an fHGB increase of 30 mg/dL, whereas the Sputnik pump induced an fHBG increase of 90 mg/dL. Taking the slightly lower circulation rate of our study into account, the HeartMate 3 induced comparable hemolysis in both studies, and the Sputnik LVAD induced comparable hemolysis compared to the HeartMate 2. Bourque et al. [39] also demonstrated a higher fHGB increase in the HeartMate 2 (103 mg/dL) compared to the HeartMate 3 (54 mg/dL) at a more clinically relevant operation condition of 5 L/min. Due to the comparable structure of the HeartMate 2 and the Sputnik pump and the similar hemolysis rates at 2 L/min, the HeartMate 3 will most likely outperform the Sputnik pump for the more clinically relevant operation condition of 5 L/min as well.

The stronger hemolysis in the Sputnik group can be explained by its higher required pump speed and its smaller blood flow gaps, which result in higher shear stress. The higher pump speed can be attributed to distinct hydraulic pump properties, which reveal typical differences between centrifugal and axial-flow LVADs [34]. The direct comparison of the pressure-flow field for Sputnik [30] and HeartMate 3 [40] confirms this argument. Furthermore, the frictional heat generated in the mechanical cup-socket bearings might also increase blood damage [7]. Cooling systems, such as Cool-Seal, applied in Evaheart (Evaheart Inc., Houston, TX, USA) have proved effective in preventing amplified hemolysis and the denaturation of blood proteins due to temperature rise in previous studies [38,41].

### 4.2. Platelet Count and Activation

PLTs adhesion to foreign surfaces, such as the blood-contacting inner surface of LVADs, is preceded by the adsorption of various plasma proteins, such as fibrinogen [42,43]. The extent of protein adsorption and subsequent PLT adhesion is strongly determined by the texture and topography of the blood-contacting surface [44]. Superhydrophobic surfaces with nanoscale topography suppress protein adsorption, while surfaces with micrometer-sized roughness increase the adsorption of proteins [45,46,47]. The titanium-textured surface of the HeartMate 3 on the inlet graft [34] thus facilitates the formation of a robust biological lining, consisting of PLTs, red blood cells, fibrin and other blood-borne materials [48]. The significant decrease in the PLT count, within the first hour of circulation in the HeartMate 3 test circuit, as well as the increase in PLT activation at 5 min, can be explained by the initiation of such a stable, biological lining on the inlet graft of the HeartMate 3. After one hour, the PLT count does not further decrease, and PLT activation decreases back to baseline levels. This indicates that the lining is already functional within the first hour of circulation and prevents further PLT activation in HeartMate 3, as PLT activation is significantly lower compared to Sputnik LVAD after 120 min.

PLT activity state in patients without thromboembolic events has been compared previously for the HeartMate 2 and HeartMate 3 LVAD [49]. In accordance with our results, the axial-flow pump (HeartMate 2) induced more PLT activation.

The time-dependent increase in PLT activation in the Sputnik group is in line with time-dependent hemolysis. Released fHGB from the destructed erythrocytes can encourage PLT activation by direct activation of PLTs via binding to glycoprotein (GP)1bα on PLTs surface or indirectly, via inhibition of nitric oxide bioactivity, resulting in decreased activation of guanylate cyclase (sGC) in PLTs, leading to enhanced activation [50]. This indirect influence mechanism of fHGB on PLT activation could be an explanation for time-dependent PLT activation, in addition to direct PLT activation related to the pump-induced shear stress [12].

### 4.3. Coagulation Activation

Increased activation of coagulation, which is sensitively indicated by elevated TAT concentrations [51], is known in patients with LVADs and was previously described in various clinical studies [15,52,53]. In accordance with previous experimental LVAD studies [54,55] utilizing an in vitro test circuit, our study likewise demonstrated a time-dependent activation of the coagulation system. Nevertheless, TAT concentrations did not significantly differ between both groups. Consequently, these findings cannot attest differences between axial or centrifugal LVAD pumps concerning the activation of coagulation.

This is in contrast to Schibilsky et al. [54] who demonstrated the superiority of a levitated centrifugal blood pump to the HeartMate 2 axial-flow pump with regard to the activation of coagulation. However, in that study, the equal flow settings resulted in even greater differences in rotational speed levels (axial-flow 9000 rpm/centrifugal-flow 2200 rpm) as compared to our present study (axial-flow (Sputnik) 5000 rpm/centrifugal-flow 3750 rpm), which could explain the studies’ contradictory findings.

### 4.4. Release of Extracellular Vesicles

EVs have been described to be increased in the circulation of LVAD patients [16,17,20,21]. However, to our knowledge, no study has yet investigated the influence of the LVAD pump itself on EV release. So far, it could also not be differentiated between EVs released due to the device and EVs that are disease related. We are the first group to compare EV generation by two different LVADs in an in vitro circuit with the blood of healthy volunteers. With our short-term in vitro model, we therefore focus solely on the device-related release of EVs from blood cells.

We could successfully isolate EVs by precipitation from baseline samples and after 6 h of circulation. When we performed Western blot analysis of our isolated EVs, we detected a significant increase in two common EV markers: Flotillin-1 and CD9 [18], which were both increased after LVAD circulation, using the Sputnik pump.

As an additional marker of EVs, we measured vesicular miRNA content, which was also significantly increased in the Sputnik samples. In pediatric patients with heart failure, it was shown that the level of six circulating miRNAs involved in the regulation of hemostatic events changed after 30 days of VAD treatment. They detected a downregulation of miR-409, which could reflect a pro-thrombotic state after VAD implantation. We have not yet analyzed the specific miRNA content from our EVs, but we are planning this analysis in future studies [56]. Taken together, our results indicate that the Sputnik pump triggers a more pronounced EV increase than the HeartMate 3 pump. This is in line with the literature, where high shear stress (<300 N/m^2^) was associated with EV formation in platelets [57]. Another study compared a roller-head pump with a centrifugal pump using pig blood and also detected higher levels of platelet EVs in the centrifugal pump, which produced more shear stress [58]. The impact of the pump on EV secretion depends on the cell type. Platelets are shown to be more susceptible to elevated shear stress environments than erythrocytes or leukocytes [59,60]. This is also supported by our experiments, as we could detect higher platelet marker (CD41) expression in Sputnik EVs than in HeartMate 3 EVs, which corresponds with our findings on PLT activation via FACS analysis, as it indicates more PLT activation in the Sputnik group. Nevertheless, as mentioned before, no differences in the general coagulation activation (TAT) between both groups were evident after 360 min of circulation. No difference in erythrocyte marker Band3/AE3 expression was detected, which leads to the conclusion that erythrocytes are not the source of EVs.

Due to the small sample volume, the physiological role of these platelet-derived EVs could not be investigated. Future experiments are needed to identify target cells and identify miRNA that could either be beneficial or detrimental. Additionally, the role of these EVs on coagulation should be studied.

### 4.5. Limitations, Outlook and Conclusions

This study was based on a miniaturized in vitro test setup for short-term blood circulation, and therefore, it has a strictly limited validity for in vivo translation or application. The test setting lacks organs and cells, which is associated with the impossibility of the regeneration or renewal of blood components, such as erythrocytes, PLTs, plasma proteins or coagulation factors. The degradation of interfering intermediate products, such as fHGB or lactate, is impossible as well, leading to their accumulation. The use of heparin as an anticoagulant, which is not used in the clinical setting, is another limitation.

The miniaturized setup, as well as the restricted pump speed to avoid speed pulsatility of the HeartMate 3, does not allow for clinically typical pump flow levels. However, blood circulates with about 1 turnover per minute in LVAD patients compared to 11.1 turnovers per minute in the ASTM F1841 test loop. Therefore, the four turnovers per minute in the miniaturized test loops appear to be a good tradeoff. The high head pressure in conjunction with low pump speed levels outside of the nominal operating range of the pumps might yield recirculation zones, leakage flows or other unfavorable flow conditions inside the LVADs specific to this operating condition. This may limit the translation of the results to some extent. However, both pumps are operated under the same conditions, and relative comparability is maintained. Therefore, future studies need to be carried out to compare the pumps under more clinically relevant operating conditions, e.g., in a standard test loop according to ASTM F1841. To allow for a head-to-head comparison, in this alternative setup, it may be necessary to implement a comparable pulsatility mode for the Sputnik pump in an experimental controller. However, due to the high filling volume of the ASTM circuit, the use of human blood may not be feasible.

Blood temperature differences between the pumps were undesired as well. These would be less pronounced inside the human body, as blood circulation would keep the inflowing blood temperature at constant levels. Furthermore, the history of the individual test pumps is unknown (they were not factory new), and only one specimen of each was used. Therefore, the results might not account for the HeartMate 3/Sputnik pumps in general. Furthermore, short-term circulation of 360 min strictly limits the validity regarding any clinical long-term application. The number of experiments (n = 7) is a further possible limitation. Nevertheless, significant differences regarding relevant parameters, such as fHGB levels or PLT count, could be shown.

In conclusion, the Sputnik LVAD device is technically less complex and more robust in comparison to the Heartmate 3 device. However, it seems to induce higher shear forces, resulting in stronger blood cell damage, respectively, activation and higher concentration of circulating extracellular vesicles. For clinical translation, further research is needed.

## Figures and Tables

**Figure 1 jcm-11-03431-f001:**
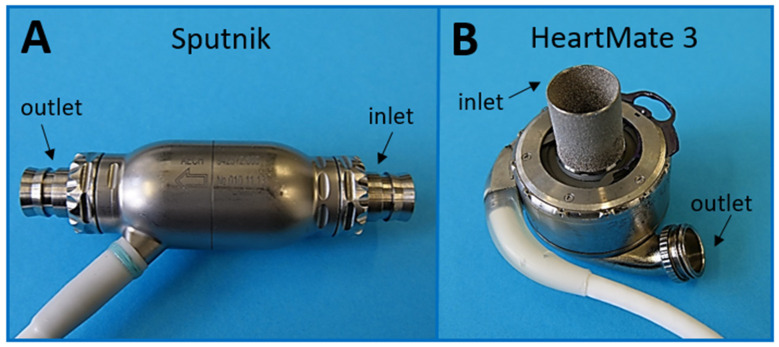
(**A**) Picture of Sputnik LVAD; (**B**) Picture of HeartMate 3 LVAD. For further details on the internal design of both pumps, refer to Mehra et al. [32].

**Figure 2 jcm-11-03431-f002:**
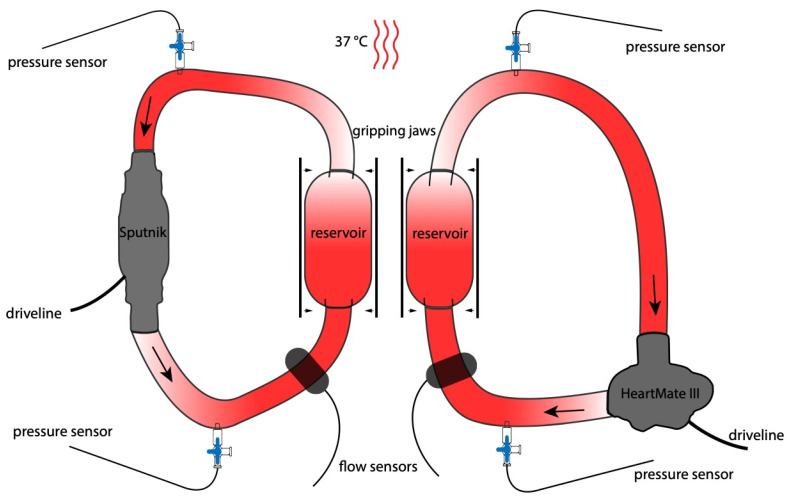
Assembly diagram of both test circuits. Large arrows: Direction of blood flow. Small arrow: Mechanical pressure on reservoirs.

**Figure 3 jcm-11-03431-f003:**
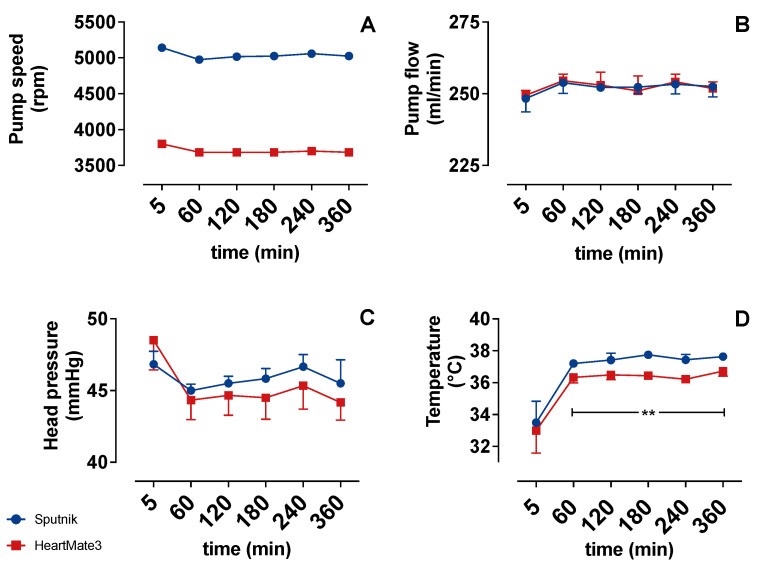
(**A**) Pump speed (rpm), (**B**) pump flow, (**C**) head pressure and (**D**) temperature over time. Significance: ** *p* < 0.01.

**Figure 4 jcm-11-03431-f004:**
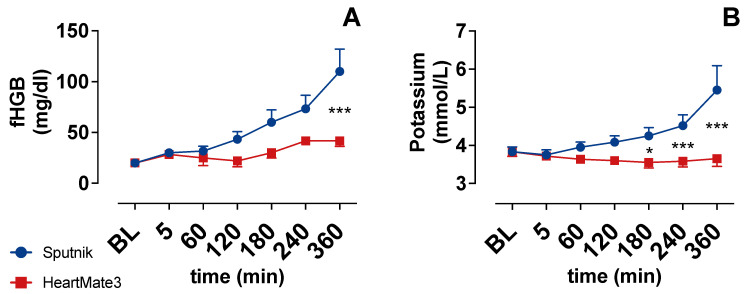
(**A**) Plasma free hemoglobin (fHGB) and (**B**) potassium over time. Baseline (BL). * *p* < 0.05; *** *p* < 0.001.

**Figure 5 jcm-11-03431-f005:**
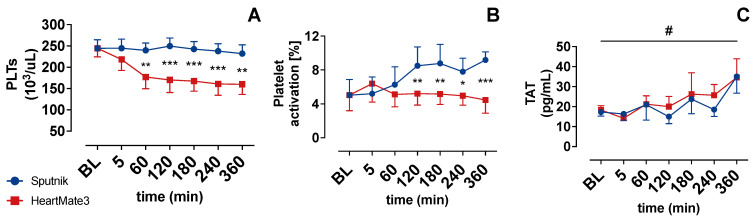
(**A**) Platelets (PLTs), (**B**) platelet activation, and (**C**) thrombin/antithrombin complex (TAT) over time. Baseline (BL). * *p* < 0.05; ** *p* < 0.01; *** *p* < 0.001; # *p* < 0.05.

**Figure 6 jcm-11-03431-f006:**
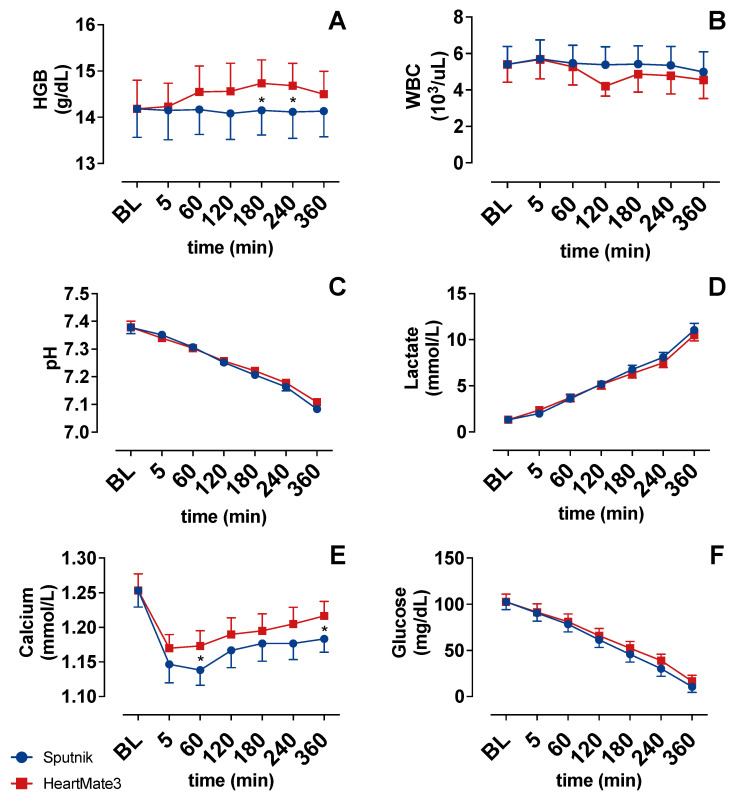
(**A**) Total hemoglobin (HGB), (**B**) white blood cell count, (**C**) pH value and (**D**) lactate concentration, (**E**) calcium concentration and (**F**) glucose concentration over time. Baseline (BL). * *p* < 0.05.

**Figure 7 jcm-11-03431-f007:**
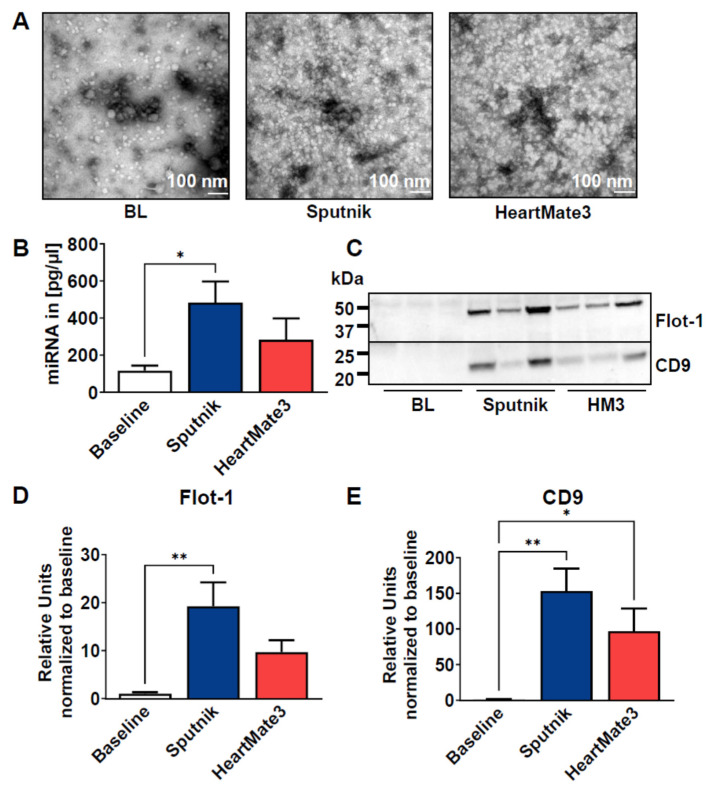
(**A**) Representative TEM images of EVs isolated by precipitation from the baseline in both Sputnik and HeartMate 3 test loops after 6 h. (**B**) MiRNA was isolated from EVs and quantified using a eukaryotic total RNA Pico chip. (**C**) Representative Western blot images of common EV markers in EVs isolated at baseline and after 6 h of circulation using the Sputnik and HeartMate 3 pumps. (**D**,**E**) Western blot bands were quantified and normalized to the corresponding total protein amount visualized in the acrylamide gels by stain-free technology and then compared to baseline. Data represent means ± SEM of six independent samples. Significance: * *p* < 0.05; ** *p* < 0.01.

**Figure 8 jcm-11-03431-f008:**
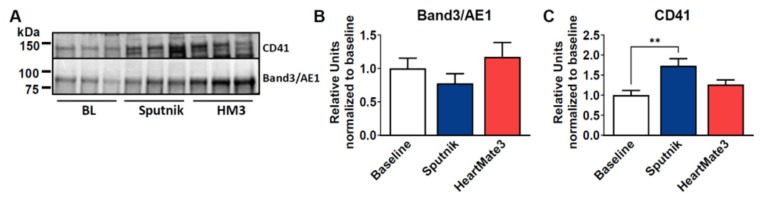
(**A**) Representative Western blot images of common EVs markers in EVs isolated at baseline and after 6h of circulation using the Sputnik or HeartMate 3 pump. (**B**,**C**) Western blot bands were quantified and normalized to the corresponding total protein amount visualized in the acrylamide gels by stain-free technology and then compared to baseline. Data represent means ± SEM of six independent samples. Significance: ** *p* < 0.01.

**Table 1 jcm-11-03431-t001:** Comparison of Sputnik and HeartMate 3 pump characteristics ([9,26,30,31,35,36]).

	Sputnik	HeartMate 3
**Pump type**	Axial-flow pump (2. generation)	Centrifugal-flow pump (3. generation)
**Bearing**	Mechanical (cup socket)	Magnetic
**Implantation location**	Extrathoracic/intrathoracic	Intrathoracic
**Patients**	>50 (2020)	>515 (2019)
**Maximum speed range**	5000 rpm–10,000 rpm	3000 rpm–9000 rpm
**Clinical speed range**	7500 rpm–8600 rpm	5000 rpm–6000 rpm
**Artificial pulse**	No	Yes
**Maximum flow**	Up to 10 L/min	Up to 10 L/min
**Weight**	246 g	200 g
**Size**	Diameter: 34 mm Length: 81 mm	Diameter: 69 mm Height: 30 mm
**Flow gap**	0.2 mm	0.5 mm–1 mm
**Material**	Titanium alloy	Titanium
**Textured surfaces**	No	Yes

## Data Availability

The datasets generated and/or analyzed during the current study are available from the corresponding author upon reasonable request.

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
