# Peer review of "Comparison of the Hemocompatibility of an Axial and a Centrifugal Left Ventricular Assist Device in an In Vitro Test Circuit"

_jcm, 2022, doi:10.3390/jcm11123431_

Round 1
Reviewer 1 Report
In this manuscript by Breuer and Bleilevens, the authors compare the hemocompatibility of an axial (Sputnik) and a centrifugal (HM 3) left ventricular assist device in an in vitro test circuit. The pumps (n=7 in each group) were operated with equal flow for six hours into identical in vitro test circuits, each filled with 75 mL heparinized human blood of the same donor. Sputnik pump induced stronger hemolysis compared to HM3.
The manuscript is well written and clear.
However, the methodology does not allow the question to be answered.
Major concerns:
- We already have human data concerning the difference of hemocompatibility between axial and centrifugal device (cf momentum 3 study and Intermacs registry)
- The design of the study has to much bias to be interpreted in clinical setting :
o The pump speeds are not in the physiological range
o Use of antiplatelet therapy?
o Use of anti Vitamin K therapy?
o Short number of pumps and duration of the study
o Not assessing the pulsatility of the HeartMate 3 is questionable as it likely plays a role in the hemocompatibility of the device. Indeed, the authors are comparing 2 devices rather than 2 technologies. If we really want to compare 2 technologies, it makes more sense to compare HM 2 and HM3 with an in vivo study.
Minor concerns:
The paragraph on extracellular vesicles brings confusion to the article. The assay method has biases as mentioned by the authors. There is no significant difference between the groups and the role of these vesicles in hemocompatibility is uncertain.
Author Response
Response to Reviewer 1 Comments
Point 1: We already have human data concerning the difference of hemocompatibility between axial and centrifugal device (cf momentum 3 study and Intermacs registry)
Answer: Thank you for this insightful comment. The Reviewers remark is correct, there is a large data set available in terms of the momentum3 study from 2016, which was recently refreshed by Kanwar et al (Center Variability in Patient Outcomes Following HeartMate 3 Implantation: An Analysis of the MOMENTUM 3 Trial - ScienceDirect). This excellent clinical trial shows the outcome of the HeartMate 2 and Heart Mate 3 pump in a large patient collective, which underlines the rationale of the authors in the herein submitted manuscript, to use the HeartMate as the comparable device to the Sputnik device (lines 84-85), which is more or less unknown and no clinical data is available so far. The authors added both studies suggested by the reviewer (momentum 3 study and the recent updates) and the following part to the introduction to clarify the intention of the manuscript (lines 86-95):
“Especially in the Asia-Pacific region, there is a high interest in lower-cost alternatives for established LVADs to offer LVAD therapy to a larger patient population [29]. One potential alternative device is the Sputnik LVAD, which is in clinical use in Russia, but data on clinical performance, or hemocompatibility properties is lacking. The aim of this study was to compare the hemocompatibility of the Sputnik LVAD with a gold standard LVAD, the HeartMate 3 in this case, in an in vitro and short-term experimental study for the first time. Therefore, human whole blood is circulated in two identical test circuits, and several parameters are determined to assess the hemocompatibility of both LVADs. However, this study solely represents a first insight on an experimental level.”
Point 2: The design of the study has too much bias to be interpreted in clinical setting:
- The pump speeds are not in the physiological range
- Use of antiplatelet therapy?
- Use of anti Vitamin K therapy?
- Short number of pumps and duration of the study
- Not assessing the pulsatility of the HeartMate 3 is questionable as it likely plays a role in the hemocompatibility of the device. Indeed, the authors are comparing 2 devices rather than 2 technologies. If we really want to compare 2 technologies, it makes more sense to compare HM 2 and HM3 with an in vivo study.
Thank you for this valuable comment. The Reviewers view is completely correct. We cannot interpret our data in terms of clinical advantages of the Sputnik device in comparison to the HeartMate3. The wording in our introduction may be misleading in this respect. Therefore, we revised the corresponding introduction part (line 98-107) completely. The usage of a rather low pump speed has been discussed in detail in lines 581-595 of the Limitations chapter. In addition, the Limitations part of the Conclusion was extended, indicating the use of heparin instead of antiplatelet and Vitamin K therapy (lines 574-575). However, the use of heparin for in vitro experiments is in accordance with the “Standard Practice for Assessment of Hemolysis in Continuous Flow Blood Pumps” (ASTM F1841). Furthermore, the small number of pumps and the short duration of the study are stated as limitations in lines 598-600 and lines 601-602, respectively. In general, we again underlined throughout the whole manuscript, that the clinical impact is “strictly limited” (lines 570; 601) and removed parts that indicate the opposite (lines 34-35; 98-107; 576-580; 610-611).
Point 3: The paragraph on extracellular vesicles brings confusion to the article. The assay method has biases as mentioned by the authors. There is no significant difference between the groups and the role of these vesicles in hemocompatibility is uncertain.
Thank you for this comment. To reduce confusion and improve reading flow we shortened the extracellular vesicles (EVs) part and moved one figure into Supplementary Materials. However, our intention was, to bring EVs into account as additional marker for shear stress and cell destruction, which could be shown by significantly increased amount of Flot-1 and CD9 positive EVs in the Sputnik loop compared to the initial measures at BL, and increased miRNA levels which are a sign of cell destruction and therewith increased shear forces in the Sputnik device. Therefore, our feeling is, that there is a rationale for having EVs in future on board as a blood cell damage marker in the LVAD application, as shown in the cited literature [16,17,20,21]. However, our results are strictly limited to experimental data and we reinforce this at the distinct parts in the discussion (lines 520-521).

Reviewer 2 Report
In the manuscript "Comparison of the hemocompatibility of an axial and a centrifugal left ventricular assist device in an in vitro test circuit" the authors investigated the effects of two different ventricular assist device on hemolysis/coagulation activation.
I wanted to congratulate the authors on this manuscript which I have read with great interest.
I only have two questions:
1)The authors extracted and quantified the miRNAs. Do the authors plan to sequence of myrinoma and to evaluate the possible targets of the miRNAs?
2) I suggest the authors to cite the article:"Variations of circulating miRNA in paediatric patients with Heart Failure supported with Ventricular Assist Device: a pilot study. Scientific reports, (2020).10(1), 5905. https://doi.org/10.1038/s41598-020-62757-7"
Author Response
Response to Reviewer 2 Comments
Point 1: The authors extracted and quantified the miRNAs. Do the authors plan to sequence of myrinoma and to evaluate the possible targets of the miRNAs?
Point 2: I suggest the authors to cite the article:"Variations of circulating miRNA in paediatric patients with Heart Failure supported with Ventricular Assist Device: a pilot study. Scientific reports, (2020).10(1), 5905. https://doi.org/10.1038/s41598-020-62757-7"
Thank you for your valuable feedback and question. We added the citation, you suggest in the discussion and answer your question as follows (lines 544-549):
“As an additional marker for EVs, we measured vesicular miRNA content which was also significantly increased in the Sputnik samples. In pediatric patients with heart failure it was shown that the level of six circulating miRNAs, involved in the regulation of hemostatic events, changed after 30 days of VAD-treatment. They detected a downregulation of miR-409, which could reflect a pro-thrombotic state after VAD-implantation [60]. We did not yet analyze the specific miRNA content from our EVs, but we are planning this analysis in future studies [56].”

Round 2
Reviewer 1 Report
The modifications made by the authors to the manuscript are adapted and allow a better understanding of the interest and the limits of this work.